# COVID-19 Vaccine Acceptance and Hesitancy among Healthcare Workers in Lusaka, Zambia; Findings and Implications for the Future

**DOI:** 10.3390/vaccines11081350

**Published:** 2023-08-09

**Authors:** Steward Mudenda, Victor Daka, Scott K. Matafwali, Phumzile Skosana, Billy Chabalenge, Moses Mukosha, Joseph O. Fadare, Ruth L. Mfune, Bwalya A. Witika, Mirriam G. Alumeta, Webrod Mufwambi, Brian Godman, Johanna C. Meyer, Angela G. Bwalya

**Affiliations:** 1Department of Pharmacy, School of Health Sciences, University of Zambia, Lusaka 10101, Zambia; moses.mukosha@unza.zm (M.M.); alumetamirriam@gmail.com (M.G.A.); webrod.mufwambi@unza.zm (W.M.); angela.bwalya@unza.zm (A.G.B.); 2Department of Public Health, Michael Chilufya Sata School of Medicine, Copperbelt University, Ndola 21692, Zambia; victordaka@cbu.ac.zm (V.D.); lindizyani@gmail.com (R.L.M.); 3Clinical Research Department, Faculty of Infectious and Tropical Diseases, London School of Hygiene & Tropical Medicine, Keppel Street, London WC1E 7HT, UK; scott.matafwali@lshtm.ac.uk; 4Department of Clinical Pharmacy, School of Pharmacy, Sefako Makgatho Health Sciences University, Pretoria 0208, South Africa; phumzile.skosana@smu.ac.za; 5Department of Medicines Control, Zambia Medicines Regulatory Authority, Lusaka 31890, Zambia; bchabalenge@zamra.co.zm; 6Department of Pharmacology and Therapeutics, Ekiti State University College of Medicine, Ado-Ekiti 362103, Nigeria; joseph.fadare@eksu.edu.ng; 7Department of Medicine, Ekiti State University Teaching Hospital, Ado-Ekiti 362103, Nigeria; 8Department of Pharmaceutical Sciences, School of Pharmacy, Sefako Makgatho Health Sciences University, Pretoria 0208, South Africa; bwalya.witika@smu.ac.za; 9Department of Public Health Pharmacy and Management, School of Pharmacy, Sefako Makgatho Health Sciences University, Pretoria 0208, South Africa; hannelie.meyer@smu.ac.za; 10Department of Pharmacoepidemiology, Strathclyde Institute of Pharmacy and Biomedical Science, University of Strathclyde, Glasgow G4 0RE, UK; 11South African Vaccination and Immunisation Centre, Sefako Makgatho Health Sciences University, Molotlegi Street, Garankuwa, Pretoria 0208, South Africa

**Keywords:** acceptance, attitudes, COVID-19 vaccines, healthcare workers, hesitancy, Zambia

## Abstract

The uptake of COVID-19 vaccines is critical to address the severe consequences of the disease. Previous studies have suggested that many healthcare workers (HCWs) are hesitant to receive the COVID-19 vaccine, further enhancing hesitancy rates within countries. COVID-19 vaccine acceptance and hesitancy levels are currently unknown among HCWs in Zambia, which is a concern given the burden of infectious diseases in the country. Consequently, this study assessed COVID-19 vaccine acceptance and hesitancy among HCWs in Lusaka, Zambia. A cross-sectional study was conducted among 240 HCWs between August and September 2022, using a semi-structured questionnaire. Multivariable analysis was used to determine the key factors associated with vaccine hesitancy among HCWs. Of the 240 HCWs who participated, 54.2% were females. A total of 72.1% of the HCWs would accept being vaccinated, while 27.9% were hesitant. Moreover, 93.3% of HCWs had positive attitudes towards COVID-19 vaccines, with medical doctors having the highest mean attitude score (82%). Encouragingly, HCWs with positive attitudes towards COVID-19 vaccines had reduced odds of being hesitant (AOR = 0.02, 95% CI: 0.01–0.11, *p* < 0.001). Overall, acceptance of the COVID-19 vaccine among HCWs in Lusaka, Zambia, was high, especially by those with positive attitudes. However, the current hesitancy among some HCWs is a concern. Consequently, there is a need to address this and encourage HCWs to fully promote vaccination programs going forward.

## 1. Introduction

The emergence of the coronavirus disease (COVID-19) and, later, its declaration as a pandemic, is among the most devastating global public health challenges in recent history [1,2]. The pandemic, alongside the consequences of the lockdown measures, resulted in a considerable burden to healthcare systems and patients, with appreciable morbidity, mortality and economic consequences, especially among low- and middle-income countries (LMICs) [3,4,5,6,7,8]. There was also an appreciable impact from COVID-19 on healthcare workers (HCWs) in terms of increased morbidity, which includes increased anxiety, depression and severe burnout, as well as increased mortality [8,9,10]. There was often inadequate personal protective equipment (PPE) at the start of the pandemic, and typically HCWs had no choice but to continue providing patient care [8]. 

A variety of preventative measures were introduced across countries to reduce the transmission and spread of COVID-19 in the absence of effective vaccines and treatments [11,12,13]. A number of treatments were proposed to treat COVID-19, which included hydroxychloroquine, lopinavir–ritonavir, remdesivir, ivermectin and dexamethasone; however only dexamethasone showed patient benefit despite the appreciable hype [14,15,16,17,18,19,20]. The public health interventions prior to the availability of potential treatments and effective vaccines included encouraging adequate handwashing, social and physical distancing, as well as the wearing of face masks [12,13,21,22,23,24]. The introduction of effective vaccines against COVID-19 further reduced the spread of the virus, as well as its impact among infected patients [21,25,26,27,28]. Unsurprisingly, HCWs were typically given first priority in receiving the COVID-19 vaccines across countries, as they were at high risk of contracting the disease, with the associated impact on their morbidity and mortality, coupled with their key role in providing care to patients with COVID-19 [8,29,30,31,32]. As such, the vaccine gave them additional confidence to treat patients with COVID-19, along with ongoing protective measures including PPE. However, the use of PPE had to be sustained for maximum protection [33]. Alongside this, continuing to adopt healthy lifestyles was important, which was a concern during the pandemic [34].

However, despite their documented effectiveness and safety across populations, hesitancy towards COVID-19 vaccines has been documented globally, with considerable concern about low acceptance rates in many countries and populations [35,36,37,38,39]. This includes Africa, with its high hesitancy rates [39,40,41,42]. Vaccine hesitancy has been defined as the delay in acceptance or refusal of vaccinations, despite the availability of vaccination services in specified areas [43]. COVID-19 vaccine hesitancy among the general public and other groups has been fueled by fear of the associated side effects and concerns regarding the effectiveness of COVID-19 vaccines in reality [39,40,44,45]. These fears are exacerbated by conspiracy theories, beliefs, myths and misinformation, fueled by misinformation on social media that was not challenged by HCWs [45,46,47,48,49,50,51,52].

HCWs can play a critical role in reducing vaccine hesitancy by offering advice and education regarding vaccinations to the community at large [10,53,54]. Activities include providing accurate information concerning the risks and benefits of COVID-19 vaccinations [54,55,56]. In a global survey of 2963 HCWs from 12 countries, 75% of the study participants knew that vaccines would control COVID-19; however, acceptance rates varied across participating countries with higher rates seen in Brazil and Malaysia, while lower rates were reported in Egypt [55]. Encouragingly, another study conducted among HCWs across 23 countries reported a vaccine hesitancy of only 15%, the lowest among physicians [57]. In addition, in their survey involving healthcare facilities in six LMICs, Baral et al. reported low levels of vaccine hesitancy, with over 90% of HCWs in healthcare facilities in Bangladesh, Liberia, Malawi and Nigeria reporting that almost all their staff had been vaccinated at the time of the survey [58]. Solís Arce et al., in their study involving 15 surveys covering 10 LMICs, also found considerable willingness by HCWs to be vaccinated, with a mean of 80.3% (median 78% and range 30.1%) [49]. High rates of over 96% of fully vaccinated HCWs have been reported among HCWs in Central and West Asia [59]. High rates of acceptance or actual vaccination have also been reported in South Africa (89% of HCWs vaccinated) [60], Lebanon (86.8% acceptance among HCWs, with 7.3% undecided) [61] and Malawi (82.5% of HCWs had received the first dose of the vaccine) [62]. 

However, lower rates of vaccine acceptance have been reported in a number of individual LMICs. In Cameroon, only 17.3% of HCWs had been vaccinated based on a recent study by Aseneh et al. (2023) [63], with low vaccination rates (26.5% of HCWs) and only a 56.2% acceptance rate in two different studies in Nigeria [63,64]. There have also been low acceptance rates of COVID-19 vaccines among HCWs in the Democratic Republic of Congo (27.7%) [65]. Low acceptance rates have also been seen in Ethiopia at 42.7% and 48.4% in two different studies [66,67], with an acceptance rate of only 57% among HCWs in Sudan [68]. An acceptance rate of only 59% among HCWs was seen in another study in South Africa [69]. 

Key factors associated with vaccine hesitancy among HCWs, particularly in LMICs, include concerns with the effectiveness and safety of the vaccines, which have been exacerbated by concerns about the short development times, the fear of getting COVID-19 from the vaccines, as well as a lack of information and misinformation [63,64,68,70].

The high rates of vaccine hesitancy seen among HCWs are a concern that needs to be urgently addressed given, as mentioned, the critical role HCWs play in helping to combat hesitancy towards COVID-19 vaccines [53,54,55]. This is important since, as mentioned, lockdown and the other measures associated with COVID-19 have caused considerable economic problems for the population, especially in LMICs, as well as negatively impacting on the care provided for other diseases [5,6,71,72,73,74,75]. Alongside this, other unintended consequences of the lockdown measures introduced to slow the spread of COVID-19 in the absence of effective vaccines included the appreciable disruption of the routine vaccination program for children across Africa [76]. This disruption has had a devastating impact on the future morbidity and mortality of these children, which cannot be readily addressed [76,77]. There have also been significant unintended consequences adversely impacting on the management of patients with non-communicable diseases (NCDs) as a result of lockdown measures, with NCDs a growing priority across Africa [75,78]. The consequences include the disruption of early diagnosis and management of patients with cancer, as well as severe effects on the routine follow-up of patients with cardiovascular disease, including diabetes [75,78,79,80]. Alongside this, there have been high rates of inappropriate prescribing of antibiotics among patients with COVID-19 across LMICs fueling antimicrobial resistance (AMR), which is already a considerable concern across Africa [81,82,83,84,85,86,87,88]. Consequently, it is vital that patients across Africa are fully vaccinated to reduce the consequences of COVID-19, as well as the unintended consequences, which have occurred as a result of a variety of measures introduced to reduce the spread of the disease in the absence of effective vaccines and treatments. 

Ensuring the necessary high levels of COVID-19 vaccination among the population will be a challenge if HCWs are concerned about the effectiveness and/or safety of the current COVID-19 vaccines [53,54,55]. We have already seen the devastating impact that misinformation regarding vaccines, including among HCWs, can have on future infection rates and this should be avoided wherever possible. The discredited study by Wakefield, published in 1998, regarding an alleged link between the MMR vaccine and autism still, to this day, has a negative impact on the uptake of the MMR vaccine across countries, resulting in increased rates of measles [89,90,91]. Alongside this, the unproven concerns that the polio vaccine causes AIDS, sterility and cancer, resulted in boycotts of this vaccine across Africa without successful counterarguments from HCWs, and in Cameroon the anti-tetanus campaign was halted due to concerns that the vaccine would make young women infertile despite convincing evidence otherwise [92,93]. These issues and concerns need to be urgently addressed by HCWs, as the continuing misinformation regarding vaccines has resulted in a reduction in parental demand for immunization for children, the highest among African countries, as well as potential negative impacts on malaria vaccination campaigns [94,95]. Such disturbing effects need to be avoided going forward with respect to the COVID-19 vaccines, especially given the considerable negative impact of the lockdown measures on routine vaccination programs for children across Africa [76,77,96]. HCWs are a critical component to address misinformation and associated vaccine hesitancy [19,53,54,55]. 

In Zambia, the Ministry of Health (MoH) launched its COVID-19 vaccination rollout on 14 April 2021 [97]. The program began with the distribution of the vaccines, which were provided through the COVAX initiative, a global collaboration aimed at ensuring equitable access to vaccines for all countries [98]. This was at no cost to citizens, in order to enhance uptake rates [98]. The initial target groups for vaccination in Zambia included HCWs, the elderly, and those with comorbidities, as they were considered the most vulnerable to severe complications from the disease [97]. The effectiveness of the vaccines, coupled with the other preventative measures, has resulted in an appreciable slowdown in new cases over the past 12 to 18 months in Zambia, with to date (18 July 2023) over 347,000 cases of COVID-19 reported since the start of the pandemic and just over 4000 deaths [99]. Despite these low mortality rates compared with a number of other countries, including Western European countries, there is a need to ensure that any vaccine hesitancy regarding COVID-19 vaccines in Zambia is kept to a minimum. Thus, limiting the potentially devastating impact that COVID-19 and its variants could have in Zambia in the future. Alongside this, limiting any ‘spillover’ effect from vaccine hesitancy with respect to the COVID-19 vaccines to other vaccination programs, as seen with the reduction in parental demand for immunization for children and malaria vaccination programs in Africa [94,95]. As mentioned, this is especially important among African countries struggling to catch up on their routine childhood vaccination programs following the considerable disruption caused by the lockdown measures at the start of the pandemic [76,77]. 

We are aware in Zambia that studies have been undertaken assessing acceptance and hesitancy toward COVID-19 vaccines among pupils [98], students [38,100] and the general population [101,102]. However, we are unaware of any published study on the attitudes and acceptance of COVID-19 vaccines and associated hesitancy among frontline HCWs in Zambia. This is imperative given their crucial role in addressing vaccine hesitancy, including misinformation, across populations and the wide variations seen in hesitancy rates across LMICs, including Africa [49,53,54,55,59,60,61,62,63,64,65,66,67,68,69,103,104]. Consequently, this study sought to address this critical information gap among this crucial population by assessing their attitudes towards and acceptance of COVID-19 vaccines, and any associated hesitancy, in two general hospitals in Lusaka, Zambia. This information gap needs to be urgently addressed given the current high rates of both infectious and non-infectious diseases across Zambia, and the need to instigate effective measures given the devastating impact of the lockdown measures on the management of NCDs and AMR [74,75,76,77,105,106,107,108]. This was the objective behind this study. As of 10 June 2023, only 61.4% of the population had received at least one dose of the vaccine [109]. 

## 2. Materials and Methods

### 2.1. Study Design, Site and Population 

This was a cross-sectional study conducted among HCWs from two purposively selected general hospitals in Lusaka, Zambia, namely Matero General Hospital and Chilenje General Hospital. These hospitals were carefully chosen for this initial study as they represent typical general hospitals in Lusaka, Zambia, and they were among the hospital sites in Zambia where COVID-19 patients were quarantined. At the time of the study, the COVID-19 vaccines were readily available at no cost to the population at these facilities. To be eligible for the study, HCWs (nurses, doctors, pharmacists, clinical officers, physiotherapists, biomedical scientists and dentists) had to provide written informed consent for participation, as required by the Ministry of Health, Zambia, and be working at one of the two selected general hospitals at the time of the data collection. 

### 2.2. Sample Size Estimation and Sampling Criteria

The required sample size was determined by using Taro Yamane’s formula, as reported by Charan and Biswas [110]. With no previous study undertaken in Zambia to assess attitudes towards COVID-19 vaccines and associated hesitancy among HCWs, we used a target population of 424 HCWs (215 from Chilenje and 209 from Matero), assuming a response distribution of 50% to give the largest sample size, with a 5% margin of error at 95% confidence, which resulted in a minimum required sample size of 206 participants. To allow for incomplete or spoiled questionnaires the minimum target sample size was increased by 10% to 227 HCWs. This translated into a minimum of 115 HCWs to be recruited from Chilenje General Hospital and 112 from Matero General Hospital. 

The participants were recruited using convenience sampling as it was the most appropriate approach to enroll HCWs, in terms of the feasibility, practicality and availability of HCWs during the COVID-19 pandemic, considering the restrictions and most HCWs working shifts [111,112]. Participant recruitment continued until the minimum required sample size at each of the hospitals was attained.

### 2.3. Data Collection Instrument and Data Collection Process

Data collection was conducted using an adapted method, including the questionnaire, from a previous study that was conducted in Thailand [113]. The resultant questionnaire was reviewed for face and content validity by an expert from the University of Zambia and the final questionnaire was, subsequently, optimized to the Zambian context. We have used this approach in previous studies conducted across different countries [98,114,115,116,117].

Following this, the questionnaire was pre-tested among 10 HCWs at Chilenje General Hospital to enhance its robustness, by verifying the simplicity of the questions and the feasibility of administering the questionnaire. HCWs who participated in the pre-testing of the questionnaire were excluded from the actual study. 

After the pre-testing, the internal consistency of the revised questionnaire was assessed by calculating the Cronbach alpha. We also used principal component analysis to identify the underlying components or factors being measured by the questions. Following these activities to enhance the robustness of the questionnaire and its validity, the final questionnaire was used to collect data on the socio-demographic characteristics, attitudes towards the COVID-19 vaccines, acceptance of the COVID-19 vaccines and factors that affect both the acceptance and hesitancy towards the COVID-19 vaccines among HCWs. The data were collected by two trained data collectors (one per facility) from August 2022 to September 2022. The participants were recruited in-person by the data collectors. The questionnaire was self-administered, to help adhere to the COVID-19 prevention measures in the participating hospitals. The questionnaires were distributed per department during working days and each participant spent an average of 20–30 min to complete the questions. Once completed, the questionnaires were placed in an envelope by the participants and returned to the data collectors during the day. Alternatively, they were submitted to the hospital reception offices for each department for collection. The questionnaire was not included in the final analysis if it was incomplete.

### 2.4. Variables 

The primary outcome of this study was vaccine hesitancy (coded as yes = 1, no = 0). We also collected data on other variables including age (measured in years from the last birthday), sex (male or female), type of HCW (doctors, nurses, pharmacists, clinical officers, physiotherapists, biomedical scientists, dentists), the presence of any health comorbidity (yes/no), whether or not the participant cared for COVID-19 patients, whether or not they worked in a high-risk environment, and the name of the facility where they were working. 

In addition, we also measured their attitudes towards COVID-19 vaccines. Each attitude was scored as one for positive responses and zero for negative responses. These scores were added together and then divided by the total score and multiplied by 100 to calculate the percentage scores. We defined positive attitudes as scores of 50% and above, similar to the published literature [118]. We also collected information on the reasons for being hesitant or accepting the COVID-19 vaccine, among the participating HCWs.

### 2.5. Statistical Analysis

The data were entered into Excel and cleaned to ensure the completeness and correctness of all variables collected. This was conducted by ensuring that all the entered responses were written in the same manner. All analyses accounted for clustering within the two hospitals using robust estimation of standard errors. The data was analyzed using descriptive statistics to examine the magnitude of vaccine hesitancy among HCWs and their basic characteristics. Mean percentage scores with standard deviations (SD) were calculated for the attitude scores. The mean difference in the overall attitude scores among the HCWs was assessed using the ANOVA test. Where appropriate, we used the Bonferroni post-hoc test to examine the pairwise comparison after the ANOVA test. Where the respondents provided reasons for COVID-19 vaccine acceptance or hesitancy, the frequencies and percentages were calculated for each reason.

We also used binary logistic regression to determine the factors associated with vaccine hesitancy. Variables with a *p*-value of less than 20% were used to build a multivariable model. The final model was arrived at after assessing the goodness of fit and possible interactions among the significant variables. Odds ratios (OR) were presented with 95% confidence intervals (CI). In the final model, we considered a *p*-value of less than 5% as statistically significant. We performed all the analyses using STATA version 17/BE (Stata Corp., College Station, TX, USA).

### 2.6. Ethical Approval

Ethical approval was granted by the University of Zambia’s Health Sciences Research Ethics Committee (UNZAHSREC), with protocol ID: 2022112301199. Clearance to collect data from the hospitals was obtained from the Lusaka District Health Board (DHO) and the institutional management.

In addition, the participants were informed verbally and in writing about the purpose of the study before being invited to participate in the study. Subsequently, written informed consent was obtained from all the participants. Participation in this study was voluntarily and the participants were free to withdraw at any point.

## 3. Results

### Background Characteristics of Study Participants

Overall, 240 HCWs responded to the survey and their characteristics are presented in Table 1. The majority of the respondents were females (54.2%), aged between 18 to 30 years old (53.3%) and were nurses (54.7%). Most respondents had no comorbidities (94.2%), provided direct care for COVID-19 patients (65.8%) and reported that their work consisted of a high risk of aerosolization (81.3%). Finally, vaccine acceptance differed by HCW profession: medical doctors (80.4%), nurses (76.3%), pharmacists (58.3%) and other HCWs (55.6%).

Overall, 72.1% (95% CI: 65.9–77.7) of respondents accepted the COVID-19 vaccines, while 27.9% (95% CI: 22.3–34.1) were vaccine hesitant. Of the respondents who provided reasons for accepting (n = 169) or being hesitant (n = 71) to vaccination against COVID-19 (Table 2), the largest proportion 96 (56.8%) accepted being vaccinated in order to prevent COVID-19 and the least 2 (1.2%) proportion accepted to be vaccinated because the vaccine was free of charge. On the other hand, 46.5% were hesitant due to concerns about the side effects of the vaccines, with 33.8% concerned about the effectiveness of the vaccines.

Table 3 shows the percentage positive attitude scores among the HCW respondents. Overall, 93.3% of HCWs surveyed showed a positive attitude towards COVID-19 vaccines. The mean (SD) attitude score was 78.6% (18.5). The mean attitude score was highest at 82.0% (17.2) among medical doctors and lowest at 69.0% (21.2) among pharmacists. Most respondents (83.8%) agreed that COVID-19 affected the country’s economy, with 82.9% recommending the COVID-19 vaccine to their patients.

Table 4 shows the factors associated with COVID-19 vaccine hesitancy among the HCWs in Zambia. From the univariable model, the factors associated with vaccine hesitancy were the occupation of the HCW, whether they ever provided direct care to COVID-19 patients, the respondents’ work consisting of a high risk of aerosolization and their attitude score. After accounting for the effect of other variables in the multivariable model, only the attitude score of the HCWs independently predicted their hesitancy toward COVID-19 vaccines. A one percentage increase in attitude score among the HCWs was associated with reduced odds of COVID-19 vaccine hesitancy (AOR = 0.02, 95% CI: 0.01–0.11, *p* < 0.001).

## 4. Discussion

We believe this is the first study to assess the attitudes, acceptance and hesitancy towards COVID-19 vaccines among the critical HCW population in Zambia. We found that 72.1% of the surveyed HCWs accepted being vaccinated while 27.9% were hesitant, with the highest rate of acceptance among medical doctors (80.4%). Alongside this, 93.3% of the HCWs had a positive attitude towards COVID-19 vaccines, with medical doctors having the highest mean attitude scores (82%). Finally, HCWs who had positive attitudes towards the COVID-19 vaccines had reduced odds of being hesitant. 

In our initial assessment, we considered the attitudes, acceptance and hesitancy towards COVID-19 vaccines among the HCWs as a whole. However, further scrutiny of our data revealed noteworthy variations among different HCWs. We discovered that medical doctors, as mentioned, exhibited a higher acceptance rate for the COVID-19 vaccines (80.4%) compared to an acceptance rate of 76.3% for nurses, 58.3% for pharmacists and only 55.6% for other HCWs. This appreciable difference might be attributed to physicians’ comprehensive understanding of the disease severity and vaccine mechanisms compared to other HCWs in Zambia. However, further research is needed before we can say anything with certainty. Furthermore, the hesitancy level varied among HCWs, with those directly involved in COVID-19 patient care exhibiting lower hesitancy. This differentiation signifies the influence of direct experience with the disease severity on vaccine acceptance, which needs to be considered in future educational and other programs to reduce hesitancy.

Encouragingly, the high acceptance rate for the COVID-19 vaccines among HCWs in Zambia is similar to a number of other studies in LMICs. This includes acceptance rates of 76.98% in China [119] and 79.6% in Libya [120]. However, this rate is higher than the rate of only 17.3% among HCWs in Cameroon and 26.5% in Nigeria [63]. There has also been low acceptance rates among HCWs in the Democratic Republic of Congo (27.7%) [65], 42.7% to 51.3% in different studies in Ethiopia [66,67,118], and 57% to 63.8% among HCWs in Sudan [68,121]. The rate in our study is also higher than 59% found in one study in South Africa [69], 54.6% to 67.7% among HCWs in India [122,123], 68.6% among HCWs in Turkey [124] and 69% in the global survey by Noushad et al. [55]. 

However, vaccine acceptance in our study was lower than that seen in Malawi, where 82.5% of HCWs had received their first dose of COVID-19 vaccines at the time of the study [62], in India where 84.1% of HCWs had been vaccinated [125], and in one study in South Africa where 89% of HCWs had been vaccinated [60]. In addition, the results are lower than those among six LMICs, where over 90% of HCWs had already been vaccinated [58], lower than 92% of HCWs in another study in Nigeria [126], and lower than those seen in Central and West Asian countries where over 96% of HCWs had already been vaccinated when the study was conducted [59]. The rate is also lower than a mean of 80.3% (median 78% and range 30.1%) among HCWs willing to be vaccinated in the study by Solís Arce et al. involving 10 LMICs [49], 85% of HCWs in the survey by Leigh et al. across 23 countries [57], and an acceptance rate of 86.8% among HCWs in Lebanon with 7.3% undecided [61]. In addition, the rate is lower than the 94.9% acceptance rate among HCWs in Singapore [127].

We believe the high acceptance for COVID-19 vaccines in our study can be attributed to the fact that HCWs were protecting themselves from the disease and its associated severe symptoms. Additionally, most HCWs in our study accepted being vaccinated because they were a high-risk population, as they provided direct patient care during the pandemic. This is similar to findings in Thailand [128] and the US [129]. In addition, some HCWs wanted to be vaccinated due to a fear of infecting their family members, as well as the vaccines being administered at no cost. This is similar to the findings from a study in Greece, where HCWs were willing to be vaccinated to protect themselves from COVID-19, as well as due to a fear of infecting their families [130]. 

Conversely, vaccine hesitancy among HCWs enrolled in our study was 27.9%. This was principally due to concerns regarding the potential side effects of the vaccines, uncertainty about their effectiveness, and low confidence about the vaccine and its source. The reasons for vaccine hesitancy are similar to other studies across LMICs [63,64,67,68,70]. Encouragingly, these rates were lower than those reported in a number of other LMICs. These include high hesitancy rates in Cameroon with only a limited number of HCWs being vaccinated [63], 72.3% hesitancy in the Democratic Republic of Congo [65], 60.1% in Sierra Leone [131], 59.8% in another study in India [132], 51.9% in Tunisia [133], 41% in one study in South Africa [69], 36% hesitancy in another study in Ethiopia [134] and 35.4% in another study in Nigeria [135]. However, the findings are similar to another study in Ethiopia with 25.5% of HCW respondents hesitant and 20.2% unwilling to recommend the vaccine [136], 28% in another study in Egypt with 51% of HCWs undecided [137] and 24.3% in French Guiana with 11% of HCWs unsure [138].

Overall, our study found that most HCWs had positive attitudes towards COVID-19 vaccines, which is encouraging, with medical doctors scoring highest on attitude scores compared to other HCWs. In addition, HCWs who had positive attitudes towards COVID-19 vaccines had reduced odds of being hesitant. This observation underscores the need to promote a positive perception of vaccines among HCWs to enhance vaccine acceptance rates, especially as our findings suggest that HCWs who have negative attitudes towards COVID-19 vaccines are more likely to be hesitant. Any hesitancy needs to be urgently addressed, especially among HCWs given their key role in promoting vaccinations for this and other infectious diseases. This is crucial to avoid the scenario seen with other vaccines, such as the MMR, polio, anti-tetanus and malaria vaccines discussed earlier [89,90,91,92,93,94,95]. Alongside this, if HCWs are unconvinced about the potential effectiveness, or lack of it, or the safety of the proposed treatments for COVID-19, including vaccines, they are unlikely to challenge misinformation. This was seen with the promotion of hydroxychloroquine and ivermectin on social media as treatments for COVID-19 in South Africa, despite no robust evidence of proven clinical benefits to these patients, which was typically unchallenged by HCWs [19]. Having said that, our findings are in line with those from other studies that found that HCWs who had negative attitudes towards COVID-19 vaccines were unlikely to accept being vaccinated themselves [118,137,139]. We have also seen HCWs willing to be suspended from work rather than be vaccinated [140].

These findings indicate that there is a relationship between attitudes and HCWs’ intentions to be vaccinated [124]. This is evidenced by the low rates of vaccinations among HCWs who had negative attitudes towards COVID-19 vaccines in this and other studies [63,64,67,68,118]. These findings must also be taken into consideration by all key stakeholders in Zambia, and beyond, when instigating activities to reduce future hesitancy in this crucial population.

In view of this, the observed vaccine hesitancy among some HCWs in our study requires urgent attention from healthcare and education authorities across Zambia. Our findings demonstrate an urgent need to increase educational activities among this critical group concerning the science of vaccines and their benefits in the fight against vaccine-preventable infectious diseases. Alongside this, any educational activities across Zambia should seek to increase awareness, knowledge and positive attitudes towards vaccinations. These educational activities should target students during university training and HCWs through continuous professional development (CPD), workshops, seminars and conferences. Additionally, there is a need to strengthen research activities on vaccines and strategies to address misinformation and myths concerning vaccinations. This will increasingly include strategies to address misinformation, especially via social media, given its increasing prominence in disseminating misinformation [19,46,48]. Alongside this, both students and HCWs in Zambia, and beyond, should be involved in basic research regarding infectious diseases and their spread, including COVID-19, and the subsequent manufacturing of vaccines, hand sanitizer, gloves and other equipment, including face masks, aprons, gowns, slip-proof footwear and ventilators, to reduce the spread of any future infectious disease and the consequences.

Furthermore, universities in Zambia and the Ministry of Health should develop context-specific and mass social media programs to rapidly disseminate accurate information about vaccines and reduce the impact of misinformation via social media. This includes new information about the effectiveness and safety of COVID-19 vaccines as it emerges. Such activities should eventually increase awareness about vaccines and address the high rates of vaccine hesitancy in Zambia that have been reported in this study. Alongside this, additional qualitative research in Zambia should be conducted to better understand the factors and reasons that contribute to vaccine hesitancy among HCWs and other key populations. The findings from such studies can subsequently be used to refine future strategies that better promote confidence and trust in vaccines in Zambia, thereby addressing vaccine hesitancy. We believe that with such initiatives, the acceptance and uptake of COVID-19 vaccines will improve, with positive consequences for other vaccines.

We are aware of a number of limitations to our study. Firstly, the study was conducted in only two hospitals in Zambia, which may not be representative of the whole country. Secondly, our study relied on a self-reported questionnaire, which could have introduced self-reporting bias. Despite these limitations, we believe that the insights provided by the appreciable number of HCWs taking part in our study on the attitudes, acceptance and hesitancy towards COVID-19 vaccines among HCWs in Zambia are valuable, providing crucial guidance to the authorities in Zambia.

## 5. Conclusions

Overall, we found high vaccine acceptance among HCWs in Zambia, especially those who had positive attitudes. HCWs who are vaccine hesitant and resistant are a public health concern and require coordinated activities to address their concerns, as they are supposed to be ambassadors of vaccine uptake. Future activities in Zambia should include greater broadcasting on the effectiveness and safety of COVID-19 vaccines as new data emerges. Alongside this, there is a greater need to rapidly provide adequate information on the safety, efficacy and the sources of vaccines, as this information emerges to address vaccine hesitancy among HCWs. As a result, the health authorities in Zambia need to be more proactive. Such activities increasingly include the proactive use of social media, as its use has been limited to date among the authorities in Zambia. Alongside this, the rapid review and upgrade to the curriculum for HCWs is needed, during their training and post-qualification. We will continue to monitor such activities in Zambia to reduce future hesitancy rates and the implications.

## Figures and Tables

**Table 1 vaccines-11-01350-t001:** Background characteristics of HCWs who were respondents.

Variable	Total, n (%)	Acceptance of COVID-19 Vaccines, n (%)	Hesitant towards COVID-19 Vaccines, n (%)
Age (years)			
18–30	128 (53.3)	89 (51.5)	39 (58.2)
31–40	48 (20.0)	32 (18.5)	16 (23.9)
Above 40	64 (26.7)	52 (30.1)	12 (17.9)
Sex			
Female	130 (54.2)	94 (54.3)	36 (53.7)
Male	110 (45.8)	79 (45.7)	31 (46.3)
Type of HCW (Profession)			
Doctor	46 (19.2)	37 (21.4)	9 (13.4)
Nurse	131 (54.7)	100 (57.8)	31 (46.3)
Pharmacist	36 (15.0)	21 (12.1)	15 (22.4)
Other ^a^	27 (11.3)	15 (8.5)	12 (17.9)
Facility name			
Chilenje hospital	120 (50.0)	83 (48.0)	37 (55.2)
Matero hospital	120 (50.0)	90 (52.0)	30 (44.8)
Comorbidity			
No	226 (94.2)	161 (93.1)	65 (97.0)
Yes	14 (5.8)	12 (6.9)	2 (3.0)
Ever provide direct care for COVID-19 patients			
No	82 (34.2)	52 (30.1)	30 (44.8)
Yes	158 (65.8)	121 (69.9)	37 (55.2)
Work consisting of a high risk of aerosolization			
No	45 (18.8)	27 (15.6)	18 (26.9)
Yes	195 (81.3)	146 (84.4)	49 (73.1)

Key: ^a^ HCWs including clinical officers, physiotherapists, biomedical scientists and dentists; HCWs = healthcare workers.

**Table 2 vaccines-11-01350-t002:** Reasons for accepting and being hesitant towards the COVID-19 vaccine among the HCWs.

Characteristic	Reasons	Frequency (n, %s)
Acceptance of COVID-19 vaccines (n = 169) 72.1% (95% CI: 65.9–77.7)	Prevent COVID-19	96 (56.8)
	High risk of being infected	51 (30.2)
	Organizational support	15 (8.9)
	Living with someone at high risk	5 (2.9)
	Free of charge	2 (1.2)
Hesitant towards COVID-19 vaccines (n = 71) 27.9% (95% CI: 22.3–34.1)	Concerns about side effects	33 (46.5)
	Concerns about efficacy	24 (33.8)
	Low confidence in the vaccine	14 (19.7)

**Table 3 vaccines-11-01350-t003:** Positive attitude ^a^ (n = 224, 93.3%) and percentage attitude scores on the COVID-19 vaccine among the healthcare workers.

Question/Statement	Total, n = 240 (%)	Medical Doctors, n = 46 (%)	Nurses, n = 131 (%)	Pharmacists, n = 36 (%)	Other ^b^, n = 27(%)	*p*-Value
COVID-19 is a severe disease	170 (70.8)	35 (76.1)	95 (72.5)	22 (61.1)	18 (66.7)	0.446
COVID-19 effects the economy	201(83.8)	43 (93.5)	106 (80.9)	26 (72.2)	26 (96.3)	0.012
COVID-19 is a preventable disease	196 (81.7)	39 (84.5)	104 (79.4)	29 (80.6)	24 (88.9)	0.632
I would recommend that my family members receive the COVID-19 vaccine	183 (76.3)	37 (80.4)	109 (83.2)	20 (55.6)	17 (63.0)	0.002
I would recommend that my patients receive the COVID-19 vaccine	199 (82.9)	41 (89.1)	115 (87.8)	25 (69.4)	18 (66.7)	0.006
Are you certain of vaccine effectiveness as part of pandemic control?	186 (77.5)	35 (76.1)	10 6(80.9)	23 (63.9)	22 (81.5)	0.172
Are you certain of vaccine effectiveness to prevent severe disease?	185 (77.1)	34 (73.9)	103 (78.6)	29 (80.6)	19 (70.4)	0.712
**Overall attitude mean scores % (SD)**						
	78.6 (18.5)	82.0 (17.2)	80.5 (17.5)	69.0 (21.2)	76.2 (18.6)	0.004

NB: ^a^ A positive attitude was defined by a score of 50% or more for the attitude questions. ^b^ Other HCWs including clinical officers, physiotherapists, biomedical scientists and dentists.

**Table 4 vaccines-11-01350-t004:** Factors associated with COVID-19 vaccine hesitancy among the HCWs.

Variable	Crude OR (95% CI)	*p*-Value	Adjusted OR (95% CI)	*p*-Value
Age				
18–30	1		1	
31–40	1.14 (0.56, 2.32)	0.715	1.36 (0.62, 3.01)	0.444
Above 40	0.53 (0.25, 1.09)	0.086	0.52 (0.23, 1.18)	0.118
Sex				
Female	1			
Male	1.02 (0.58, 1.80)	0.933		
Type of HCW (profession)				
Doctor	1		1	
Nurse	1.27 (0.55, 2.93)	0.568	1.25 (0.50, 3.08)	0.635
Pharmacist	2.94 (1.10, 7.86)	0.032	1.58 (0.52, 4.79)	0.099
Other	3.29 (1.15, 9.42)	0.027	2.64 (0.83, 8.36)	0.422
Facility name				
Chilenje hospital	1			
Matero hospital	0.75 (0.42, 1.32)	0.314		
Comorbidity				
No	1			
Yes	0.41 (0.09, 1.90)	0.255		
Ever provide direct care to COVID-19 patients				
No	1		1	
Yes	0.53 (0.30, 0.95)	0.032	0.78 (0.41, 1.51)	0.468
Work consisting of a high risk of aerosolization				
No	1		1	
Yes	0.50 (0.26, 0.99)	0.047	0.67 (0.31, 1.47)	0.319
Attitude score	0.02 (0.01, 0.09)	<0.001	0.02 (0.01, 0.11)	<0.001

NB: HCW= Healthcare worker; OR = Odds ratio

## Data Availability

Additional data are available on reasonable request from the corresponding authors.

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
