# Peer review of "COVID-19 Vaccine Acceptance and Hesitancy among Healthcare Workers in Lusaka, Zambia; Findings and Implications for the Future"

_vaccines, 2023, doi:10.3390/vaccines11081350_

Round 1

Reviewer 1 Report

The manuscript (ID: vaccines-2506092) aimed to explore the COVID-19 vaccine acceptance and hesitancy among healthcare workers in Zambia.   

The general impression is that the authors of the paper know how to write a scientific paper, but that they failed to plan the study better (taking into account that a cross-sectional study design was applied in this research, at one time-study point, and that the authors - 14 authors of this paper - passed lightly over the fact that the sample size is unacceptably small compared to the estimate obtained by calculating the sample size, and that it can only be assumed that a self-filling questionnaire was applied, although the paper does not state anywhere how the questionnaire was filled out and in what way the questionnaire was distributed).   

The unacceptably small sample size led to the fact that in this paper adequate assessment of the impact of the observed characteristics of the respondents on the COVID-19 vaccine acceptance and hesitancy among healthcare workers in Zambia was not achieved.         

Comments: 

  • Lines 35-36: Specify the aim of this work in the Abstract.   
  • The Introduction section presents the circumstances that indicate the importance of the topic investigated by this study.     
  • Lines 49-115: This text has no direct relation to vaccination against COVID-19 in Zambia. Enter information that will familiarize the readers of the environment in which health workers in Zambia work during the COVID-19 pandemic: provide the latest data on the number of patients and deaths from COVID-19 in Zambia, whether there were and how many cases and deaths from COVID-19 among HCWs in Zambia. Provide such data for other countries, either in the neighboring countries or across the world. It is necessary to reconstruct the Introduction section by entering the necessary data, but taking into account that this does not affect the length of the Introduction.   
  • Lines 129-132: Explain this sentence listed after the goal of this paper.
  • State and explain the reasons why these 2 hospitals were, I quote `purposively selected` (on Lines 135-135) and `carefully chosen` (on Line 137), for this study.
  • Before starting this study, did the authors know the number of HCWs employed in these 2 hospitals?    
  • Please indicate the total number of HCWs employed in these two hospitals. 
  • Lines 148-153: Indicate whether the method of questionnaire self-completion or the interview method was used in the collection of data in this paper. 
  • Lines 166-170: In this subsection, list all dependent and independent variables described in this manuscript.   
  • Lines 189-190: Provide reasons for non-response for the 60 of HCWs who were administered the questionnaire in this study. 
  • Lines 196-198: For `Acceptance of COVID-19 vaccines` and `Hesitant towards COVID-19 vaccines` determine the statistical significance of the distribution of HCWs according to all the mentioned characteristics. 
  • Lines 208-213: Describe all variables where statistical significance was reached and emphasize for which HCWs it was (highlight both maximum and minimum values). 
  • The Discussion section as a whole shows the author's ability to make good use of the relevant literature. However, the main shortcoming in the Discussion section is to observe and discuss the sample in this study as a whole, that is, all HCWs together. Notes: 
    • Discuss the differences among HCWs in this study (regarding `Reasons for accepting and being hesitant towards the COVID-19 vaccine among HCWs.`, `Positive attitude and percentage attitude scores of COVID-19 vaccine among healthcare workers`, and `Predictors of COVID-19 vaccine hesitancy among HCWs').   
    • Lines 272-286:  The text is written too generally. Discuss the issue by targeting and specifying the problem, and you yourself indicated that in the results you presented on Table 3 and Table 4. 
  • Lines 298-304: The limitations of this work are neither accurately enumerated (and there are many) nor adequately discussed. Accordingly, completely reconstruct the paragraph on the limitations of this paper.  

The quality of English language is appropriate. 

Author Response

Comments and Suggestions for Authors

The manuscript (ID: vaccines-2506092) aimed to explore the COVID-19 vaccine acceptance and hesitancy among healthcare workers in Zambia.   

The general impression is that the authors of the paper know how to write a scientific paper, but that they failed to plan the study better (taking into account that a cross-sectional study design was applied in this research, at one time-study point, and that the authors - 14 authors of this paper - passed lightly over the fact that the sample size is unacceptably small compared to the estimate obtained by calculating the sample size, and that it can only be assumed that a self-filling questionnaire was applied, although the paper does not state anywhere how the questionnaire was filled out and in what way the questionnaire was distributed).   

The unacceptably small sample size led to the fact that in this paper adequate assessment of the impact of the observed characteristics of the respondents on the COVID-19 vaccine acceptance and hesitancy among healthcare workers in Zambia was not achieved.         

Author Comments: Thank you for these helpful comments. However, we have now upgraded the Methodology to give more details as why we just chose 2 important sites for this pilot study rather than include HCWs from across Zambia. In addition, why there were challenges to recruiting HCWs in the current pandemic. Overall, we believe a sample size of 240 participants from two facilities was a good sample size for the pilot, and is above suggested figures, although participant rates could have been higher. Consequently, we believe that in times of COVID-19 with associated restrictions, and emphasis on adhering to social and physical distancing, we believe the sample size was adequate for this study, and hope this is acceptable to you.

Comments: 

  • Lines 35-36: Specify the aim of this work in the Abstract.   

Author Comments: Thank you, we added the aim in the abstract.

2) The Introduction section presents the circumstances that indicate the importance of the topic investigated by this study.     

Author Comments: Thank you for this.

3) Lines 49-115: This text has no direct relation to vaccination against COVID-19 in Zambia. Enter the information that will familiarize the readers with the environment in which health workers in Zambia work during the COVID-19 pandemic: provide the latest data on the number of patients and deaths from COVID-19 in Zambia, whether there were and how many cases and deaths from COVID-19 among HCWs in Zambia. Provide such data for other countries, either in neighbouring countries or across the world. It is necessary to reconstruct the Introduction section by entering the necessary data, but taking into account that this does not affect the length of the Introduction.   

Author Comments: Thank you for this. We have added some details into the revised manuscript. However, there were no up-to-date figures for HCWs in Zambia and neighbouring countries. Given though ongoing hesitancy regarding COVID-19 vaccines across Africa, as well as considerable issues with the implications of lockdown activities on other diseases as well as routine vaccination programmes across Africa, and the real concerns with COVID-19 vaccine hesitancy worldwide as well as hesitancy with other vaccines base on false information, we thought it prudent to lay out the Introduction in this way. We have further built on this in the Discussion. We trust this is acceptable to you as concerns with vaccine hesitancy generally can be devastating and HCWs are key to help address this. This was the whole premise of the paper.

4) Lines 129-132: Explain this sentence listed after the goal of this paper.

Author Comments: Thank you – now updated. We hope this is now OK.

5) State and explain the reasons why these 2 hospitals were, I quote `purposively selected` (on Lines 135-135) and `carefully chosen` (on Line 137), for this study.

Author Comments: Thank you, we chose these hospitals because they were sites where COVID-19 patients were quarantined. Hence, we desired to understand the vaccine acceptance and hesitancy among HCWs at these two hospitals. Additionally, COVID-19 vaccines were available in these two facilities. We have now added these details into the revised paper, and trust this is now OK.

6) Before starting this study, did the authors know the number of HCWs employed in these 2 hospitals?    

Author Comments: Dear reviewer, thank you very much for this comment. Yes, we did know the population, which was used in our study design (now revised).

7) Please indicate the total number of HCWs employed in these two hospitals. 

Author Comments: Thank you, we have now included this in the revised manuscript.

8) Lines 148-153: Indicate whether the method of questionnaire self-completion or the interview method was used in the collection of data in this paper. 

Author Comments: Thank you, to minimise contact with participants and adhere to the COVID-19 prevention measures, the questionnaires were self-completed by the participants. We have now added these details into the revised paper.

9) Lines 166-170: In this subsection, list all dependent and independent variables described in this manuscript.   

Author Comments: Thank you for this suggestion. We have since added the variables to this section and hope  this is now acceptable.

10) Lines 189-190: Provide reasons for non-response for the 60 of HCWs who were administered the questionnaire in this study. 

Author comments: Thank you, after reaching our sample size, we went back to collect the other questionnaires, however, due to shift changes, we could not collect others, and others remained unfilled, although this did not affect our required minimum sample size. Some of the health workers did not return the questionnaires due to unavailability after a change of shift, some proceeded to other work assignments and leave the facility while a few declined to participate. We have now included these details in the revised manuscript.

11) Lines 196-198: For `Acceptance of COVID-19 vaccines` and `Hesitant towards COVID-19 vaccines` determine the statistical significance of the distribution of HCWs according to all the mentioned characteristics. 

Author Comments: We appreciate this suggestion from the reviewer. However, we do not think this will add much value to the paper as these are basic univariate statistics which we typically do not use to make conclusive decisions. We would however like to draw your attention to the multivariable model, which has both univariable (will show what the reviewer is requesting in table one) and multivariable results that we feel have conclusively taken care of independence of characteristics to predict vaccine hesitancy, which was our primary outcome of interest. We hope this is acceptable to you.

12) Lines 208-213: Describe all variables where statistical significance was reached and emphasize for which HCWs it was (highlight both maximum and minimum values). 

Author Comments: We thank the reviewer for the proposed addition. We have since adjusted the narrative to highlight both the highest and minimum values, albeit the significance part as explained in comment 11. We hope this is now OK.

13) The Discussion section as a whole shows the author's ability to make good use of the relevant literature. However, the main shortcoming in the Discussion section is to observe and discuss the sample in this study as a whole, that is, all HCWs together. 

Author Comments: Thank you for this insightful comment. We understand that a breakdown of our results by specific roles of HCWs (nurses, doctors, lab technicians, etc.) could provide additional granularity to our findings. We revised the Discussion section accordingly and have addressed the differences and similarities in vaccine acceptance and hesitancy among the different roles of HCWs involved in our study. We hope this is now OK.

14) Discuss the differences among HCWs in this study (regarding `Reasons for accepting and being hesitant towards the COVID-19 vaccine among HCWs.`, `Positive attitude and percentage attitude scores of COVID-19 vaccine among healthcare workers`, and `Predictors of COVID-19 vaccine hesitancy among HCWs').   

Author Comments: We appreciate your suggestion. We agree that the differences among HCWs regarding their attitudes, acceptance, and hesitancy towards COVID-19 vaccines could further enhance our understanding of the situation in Zambia. In the revised manuscript, we have included a detailed discussion of how the various roles of HCWs in our study have influenced their perspectives on the COVID-19 vaccines, and hope this is now acceptable.

15) Lines 272-286:  The text is written too generally. Discuss the issue by targeting and specifying the problem, and you yourself indicated that in the results you presented in Table 3 and Table 4. 

Author Comments: Thank you for bringing this to our attention. We revised this section to include specific details from our findings, as shown in Tables 3 and 4, to provide a more focused discussion and address the issue in a more targeted manner. We hope this is now OK.

16) Lines 298-304: The limitations of this work are neither accurately enumerated (and there are many) nor adequately discussed. Accordingly, completely reconstruct the paragraph on the limitations of this paper.  

Author Comments: Author Comments: We appreciate this and we have now refined this section. We hope this is now acceptable.

Comments on the Quality of English Language - The quality of the English language is appropriate. 

Author Comments: Thank you for this – appreciated!

Reviewer 2 Report

Comments

This important study evaluates the COVID-2019 vaccine acceptance and hesitant levels among healthcare workers in Zambia. The manuscript, however, lacks some important information in the main text. Here are some points I would like the authors to consider to further highlight the

contribution of the study.

1.     The introduction section of the article is too long, please shorten it and just emphasize the main research question and purpose.

2.     In the methodology section of the article, the sample size is calculated to be 385, but the actual sample size of the article is only 240, which is a big kind of flaw in the article, and most likely unrepresentative.

3.     Manuscript requires ethical review information.

4.     In the methodology section of the article, the sample size is calculated to be 385, but the actual sample size of the article is only 240, which is a big kind of flaw in the article, and most likely unrepresentative.

None.

Author Response

Comments

This important study evaluates the COVID-2019 vaccine acceptance and hesitant levels among healthcare workers in Zambia. The manuscript, however, lacks some important information in the main text. Here are some points I would like the authors to consider to further highlight the

contribution of the study.

Author comments: Thank you for these comments – we hope we have adequately addressed the issues you have raised.

  1. The introduction section of the article is too long, please shorten it and just emphasize the main research question and purpose.

Author Comments: Thank you for this comment. We have now updated the Introduction in line with comments from other Reviewers. This included the fact that vaccine hesitancy due to misinformation has often not been adequately addressed by key HCWs – even worse misinformation has been endorsed by them (as we have seen with e.g. the administration of hydroxychloroquine endorsed in guidelines for the treatment of patients with COVID-19 despite no evidence of its effectiveness). Such activities by HCWs have devastating consequences not only for the infectious disease in question (as seen with continuing hesitancy towards the MMR vaccine even following the discreditation of Wakefield’s initial study) but also has spill-over effects to other vaccines (which we comment on). Consequently, the critical need to address any concerns regarding the effectiveness/ safety of the COVID-19 vaccines among HCWs. This is even more important in Africa as the routine vaccination programme was severely curtailed following lockdown measures with devastating consequences to children. We hope this is now clear and acceptable to you. We have further mentioned these concerns in the Discussion. 

  1. In the methodology section of the article, the sample size is calculated to be 385, but the actual sample size of the article is only 240, which is a big kind of flaw in the article, and most likely unrepresentative.

Author Comments: Thank you for this comment. Sorry, we have corrected this in the methods of the revised manuscript. We used the total population from the 2 study sites (we adjusted our sample size calculation to the finite population of 424 HCWs) and found a minimum of 227 required participants. However, we distributed more questionnaires due to expected off-work (rests) and working in shifts among HCWs during the pandemic. We hope this is now OK.

  1. Manuscript requires ethical review information.

Author Comments: Thank you for this observation. We initially followed the journal format in which the ethical review information was included after the ‘Conclusion’ section. However, we have now also added this in information at the end of the Methods section. We trust this is now acceptable.

  1. In the methodology section of the article, the sample size is calculated to be 385, but the actual sample size of the article is only 240, which is a big kind of flaw in the article, and most likely unrepresentative.

Author Comments: Thank you for this observation. We revised this section to highlight that for this initial study, we just used 2 hospitals that were carefully chosen (as discussed in the revised Methods section). As a result, we required a minimum of 227 HCWs based on the total population of HCWs at the 2 study sites. Hence, we adjusted our sample size calculation to this updated population. We hope this is now fine.

Comments on the Quality of English Language. None.

Reviewer 3 Report

Thank you for sharing your article on vaccine acceptance and hesitancy among Zambian healthcare workers. Here some suggested edits and comments that could help to improve your article:

L68: Vaccine hesitancy could be one reason for low vaccination rates. Please consider rephrasing for more clarity.

L74: Is the term "currently" still up-to-date? Please consider rephrasing for more clarity.

L76: Does "risks and benefits of vaccinations" refer to any vaccines or is this statement still in the context of COVID-19 vaccination? Please revise as appropriate.

L99: Do you mean prescribing antibiotics to treat COVID-19 or co-infections occurring with COVID-19? 

L136: What was the prevalence of COVID-19 at the time of study conduct in Zambia? Did you assess whether this could have had an impact on the responses given by the HCWs enrolled? Did you attempt to conduct your study at different time points during the COVID-19 epidemic in Zambia?

L167-168: How do you define "correct response" versus "wrong response"; please clarify in your manuscript. L169: Likewise, it is not fully clear what you mean by "positive attitudes". L166-170: Is the variable "attitude" what you present later as "attitude score"? This needs more clarification in your manuscript. Table 3: What is meant by "overall score" and how did you generate this. The section "2.5 statistical analysis" is a very suitable section to better explain the calculations and analyses you performed. 

L172: Please provide more detailed information on data entry as well as checking for data completeness; how did you check for data correctness and, more importantly, what measures did you undertake in case data were not correct?

L189: Why 240 HCWs only? Your original sample size was 385. You stated that you recruited 300 HCWs only (minus the pilot testing).

L201: Which vaccine was administered in Zambia and could this have had an impact on vaccine hesitancy?

L222: Which "other variables" and how did you decide on their selection?

See above.

Author Response

Comments and Suggestions for Authors

Thank you for sharing your article on vaccine acceptance and hesitancy among Zambian healthcare workers. Here are some suggested edits and comments that could help to improve your article:

1) L68: Vaccine hesitancy could be one reason for low vaccination rates. Please consider rephrasing for more clarity.

Author Comments: Thank you, we have now revised it.

2) L74: Is the term "currently" still up-to-date? Please consider rephrasing for more clarity.

Author Comments: Thank you, we have deleted the word ‘currently’.

3) L76: Does "risks and benefits of vaccinations" refer to any vaccines or is this statement still in the context of COVID-19 vaccination? Please revise as appropriate.

Author Comments: Thank you. We started with a foundation of all vaccines to emphasize the destructive impact of misinformation regarding vaccines, e.g. the discredited misinformation regarding MMR vaccines still has a negative impact today . In this sentence, we meant COVID-19 vaccinations. We have revised and added the word ‘COVID-19’, and trust this is now OK.

4) L99: Do you mean prescribing antibiotics to treat COVID-19 or co-infections occurring with COVID-19? 

Author Comments: Thank you for this observation. In most African countries, including non-African countries, there has been inappropriate use of antibiotics including in suspected and confirmed COVID-19 patients, even those with flu and felt it was COVID-19. These high levels of inappropriate used have fuelled AMR certainly in LMICs, an already known problem. We hope this is now clear.

5) L136: What was the prevalence of COVID-19 at the time of the study conducted in Zambia? Did you assess whether this could have had an impact on the responses given by the HCWs enrolled? Did you attempt to conduct your study at different time points during the COVID-19 epidemic in Zambia?

Author Comments: Thank you very much for these observations. In 2022, the prevalence of COVID-19 had reduced in Zambia. We are aware that the period the study was conducted could have affected the responses given. Additionally, we conducted a cross-sectional study which was not longitudinal in nature. We will seek to undertake longitudinal and qualitative studies in the future.  

6) L167-168: How do you define "correct response" versus "wrong response"; please clarify in your manuscript. L169: Likewise, it is not fully clear what you mean by "positive attitudes".

Author Comments: Thank you for pointing this out. We have since reworded the explanation under the variables section to state that positive response were coded as one with negative responses coded as 0. A positive attitude was derived by categorising the continuous scale (see the response on the next query - 7).

7) L166-170: Is the variable "attitude" what you present later as "attitude score"? This needs more clarification in your manuscript. Table 3: What is meant by "overall score" and how did you generate this? The section "2.5 statistical analysis" is a very suitable section to better explain the calculations and analyses you performed. 

Author comments: Thank you for this query. We have since reworked the flow for clarity and added an explanation on how attitude was scored first on a continuous scale (positive response were0 coded as one and negative response coded as 0) and responses summed to obtain overall attitude scores. The final scores were further categorised as >=50% score positive attitude and less than 50% as a negative attitude. This scoring system was based on similar studies to make it easier for us to compare our findings to similar studies.

8) L172: Please provide more detailed information on data entry as well as checking for data completeness; how did you check for data correctness and, more importantly, what measures did you undertake in case data were not correct?

Author comment: Thank you for this comment. From data entry to analysis, researchers checked that the entered data was correct. For instance, in statistical software, Female, female, and FEMALE may be entered wrongly like this. So, after data entry, we ensured that all responses were correctly entered. We used a filter button in the Excel sheet to check that all the responses were well-entered with no mistakes. We have added a sentence in the data entry and analysis section for clarification, and hope this is now OK.  

9) L189: Why 240 HCWs only? Your original sample size was 385. You stated that you recruited 300 HCWs only (minus the pilot testing).

Author Comments:  Thank you for this observation. However, we have now rectified this mistake. Our final sample size was corrected for our finite population, i.e. HCWs in the two target hospitals, which resulted in a sample size of 227 HCWs. We have now corrected this and we hope it is fine now.  

10) L201: Which vaccine was administered in Zambia and could this have had an impact on vaccine hesitancy?

Author Comments: Thank you, AstraZeneca was the first vaccine to be administered in Zambia. But later, we received Johnson and Johnson, Sinopharm and Pfizer. So, we believe there was a wider choice regarding the type of vaccines to receive among priority populations like HCWs.

11) L222: Which "other variables" and how did you decide on their selection?

Author Comments: We have updated the variables section, and these variables were chosen based on the researcher’s experience in the field and also the published literature.

12) Comments on the Quality of English Language - See above.

Author Comments: Thank you for this. We have now been through the manuscript with the help of one of the co-authors who has a native English speaker over 500 publications in peer-reviewed Journals. We trust this is now acceptable.

Round 2

Reviewer 1 Report

Thank you for the opportunity to re-review manuscript ID: vaccines-2506092. Overall, the authors submitted a version of the manuscript for re-review in which significant corrections were made: the revised manuscript is satisfactorily clear and informative for the topic it describes. The authors satisfactorily answered all my comments and provided appropriate explanations.  

I thank the authors.  

My recommendation for the revised version of this manuscript is: accept in present form.  

But there is one suggestion for the authors, since they have incorporated a different way/formula for calculating the sample size in the revised version of the paper. Namely, it is necessary to check whether an omission has been made and to correct that omission in such a way as to harmonize the information on the same issue that is presented in the following 3 places in the text of the revised paper:  

Lines 173-179:

`The sample size was determined by using Taro Yamane's formula as reported by Charan and Biswas [75]. With no previous study undertaken in Zambia to assess attitudes towards COVID-19 vaccines and associated hesitancy among HCWs, we used a population of 424 HCWs (215 from Chilenje and 209 from Matero) and a margin of error of 5% to estimate the required sample size. This resulted in a minimum sample size of 206 participants to be recruited. Taking into consideration a 10% non-response rate, the final sample size was estimated at 227 HCWs.`    

Line 241:

`Overall, 240 HCWs responded to the survey giving a response rate of 80%, and their ...`.   

Lines 381-382:

`Second, despite our best efforts, we were unable to reach the suggested number of participants, limiting the statistical power of our analysis.`    

The quality of English language is appropriate. 

Author Response

Comments and Suggestions for Authors

Thank you for the opportunity to re-review manuscript ID: vaccines-2506092. Overall, the authors submitted a version of the manuscript for re-review in which significant corrections were made: the revised manuscript is satisfactorily clear and informative for the topic it describes. The authors satisfactorily answered all my comments and provided appropriate explanations.  

I thank the authors.  

Author comments: Thank you for your kind words – appreciated!

My recommendation for the revised version of this manuscript is: accept in present form.  

But there is one suggestion for the authors, since they have incorporated a different way/formula for calculating the sample size in the revised version of the paper. Namely, it is necessary to check whether an omission has been made and to correct that omission in such a way as to harmonize the information on the same issue that is presented in the following 3 places in the text of the revised paper:  

Lines 173-179:

`The sample size was determined by using Taro Yamane's formula as reported by Charan and Biswas [75]. With no previous study undertaken in Zambia to assess attitudes towards COVID-19 vaccines and associated hesitancy among HCWs, we used a population of 424 HCWs (215 from Chilenje and 209 from Matero) and a margin of error of 5% to estimate the required sample size. This resulted in a minimum sample size of 206 participants to be recruited. Taking into consideration a 10% non-response rate, the final sample size was estimated at 227 HCWs.`    

Line 241:

`Overall, 240 HCWs responded to the survey giving a response rate of 80%, and their ...`.   

Lines 381-382:

`Second, despite our best efforts, we were unable to reach the suggested number of participants, limiting the statistical power of our analysis.`  

Author comments: Thank you for this. We have noted this inconsistency – also highlighted by the Editor – and have further addressed this discrepancy in the revised paper – emphasizing that we met the minimum number from these 2 hospitals - building on our previous re-submission.

Reviewer 2 Report

None.

Author Response

Comments and Suggestions for Authors

None.

Author comments: Thank you for this. Following comments from the Editor we have further updated the Methodology section, and trust this is now OK.

Reviewer 3 Report

Thank you for sharing the revised manuscript. All my suggested edits and comments were addressed sufficiently. 

Author Response

Comments and Suggestions for Authors

Thank you for sharing the revised manuscript. All my suggested edits and comments were addressed sufficiently. 

Author comments: Thank you – appreciated!
